# Unlocking the Potential of Permeable Pavements in Practice: A Large-Scale Field Study of Performance Factors of Permeable Pavements in The Netherlands

Ted Isis Elize Veldkamp [1,*], Floris Cornelis Boogaard [2,3,4] and Jeroen Kluck [1,5]

1. Centre of Expertise Urban Technology, Amsterdam University of Applied Sciences, Weesperzijde 190, 1097 DZ Amsterdam, The Netherlands; j.kluck@hva.nl
2. NoorderRuimte, Centre of Applied Research and Innovation on Area Development, Hanze University of Applied Sciences, Zernikeplein 7, P.O. Box 3037, 9701 DA Groningen, The Netherlands; f.c.boogaard@pl.hanze.nl
3. Global Center on Adaptation, Energy Academy Europe, Nijenborgh 6, 9747 AG Groningen, The Netherlands
4. Deltares, Postbus 177, 2600 MH Delft, The Netherlands
5. TAUW B.V., Handelskade 37, Postbus 133, 7400 AC Deventer, The Netherlands
* Correspondence: t.i.e.veldkamp@hva.nl

**Abstract:** Infiltrating pavements are potentially effective climate adaptation measures to counteract arising challenges related to flooding and drought in urban areas. However, they are susceptible to clogging causing premature degradation. As part of the Dutch Delta Plan, Dutch municipalities were encouraged to put infiltrating pavements into practice. Disappointing experiences made a significant number of municipalities decide, however, to stop further implementation. A need existed to better understand how infiltrating pavements function in practice. Through 81 full-scale infiltration tests, we investigated the performance of infiltrating pavements in practice. Most pavements function well above Dutch and international standards. However, variation was found to be high. Infiltration rates decrease over time. Age alone, however, is not a sufficient explanatory factor. Other factors, such as environmental or system characteristics, are of influence here. Maintenance can play a major role in preserving/improving the performance of infiltrating pavements in practice. While our results provide the first indication of the functioning of infiltrating pavement in practice, only with multi-year measurements following a strict monitoring protocol can the longer-term effects of environmental factors and maintenance actually be determined, providing the basis for the development of an optimal maintenance schedule and associated cost–benefit assessments to the added value of this type of climate adaptation.

**Keywords:** permeable pavements; stormwater harvesting; urban water management; climate change; climate extremes; hydrological field experiments; hydraulic performance; maintenance; full-scale infiltration tests

## 1. Introduction

Climate change is putting increasing pressure on the living environment and the urban water system of cities [1,2]. One of the ways to make cities more climate-proof is to implement a sustainable urban water drainage system [3,4], i.e., to ensure that more rainwater infiltrates the soil, and less rainwater needs to be discharged through the sewer system. This limits problems with flooding due to extreme precipitation as well as the impacts of drought [5–9]. Infiltrating pavements can be of help here, especially in areas with a high density of roads and limited space for the implementation of additional green [10–13]. Hence, several types of infiltrating pavements have been developed and are being brought to the market as an alternative to rainwater sewers to retain and store rainwater [3]. Here, a distinction can be made between impermeable concrete interlocking pavers (IPCIPs, i.e., concrete pavers with wide joints or apertures) and porous concrete interlocking pavers

(PCIPs, i.e., porous pavers with or without wide joints) [3]. Figure 1 shows a typical set-up of a Dutch infiltrating pavement structure (PCIP/IPCIP) with corresponding system characteristics. While infiltrating pavements have a clear beneficial potential to society, they are susceptible to physical and biological clogging problems causing premature degradation and serviceability challenges [14–21].

Coping with the impacts of climate extremes in an urbanized context is also an increasing challenge in the Netherlands [22,23]. Hence, The Dutch Delta Plan to tackle the impacts of climate change required all Dutch municipalities from 2017 onward to take the impacts of climate change explicitly into account when redesigning their cities [24,25]. While various municipalities had already started piloting the implementation of infiltrating pavements in practice, disappointing experiences considering the infiltration rate and uncertainties with regards to the maintenance of infiltrating pavements led a significant number of municipalities to decide to strongly reduce or completely stop further implementation of permeable pavement in practice [26–28]. As such decisions often appeared to be based on a limited number of pilots, without distinctions being made for the different types of infiltrating pavements, nor with attention devoted to the potential (positive) impact of maintenance, a clear need exists for more insight into the functioning of infiltrating pavements in practice and the effects of maintenance [29].

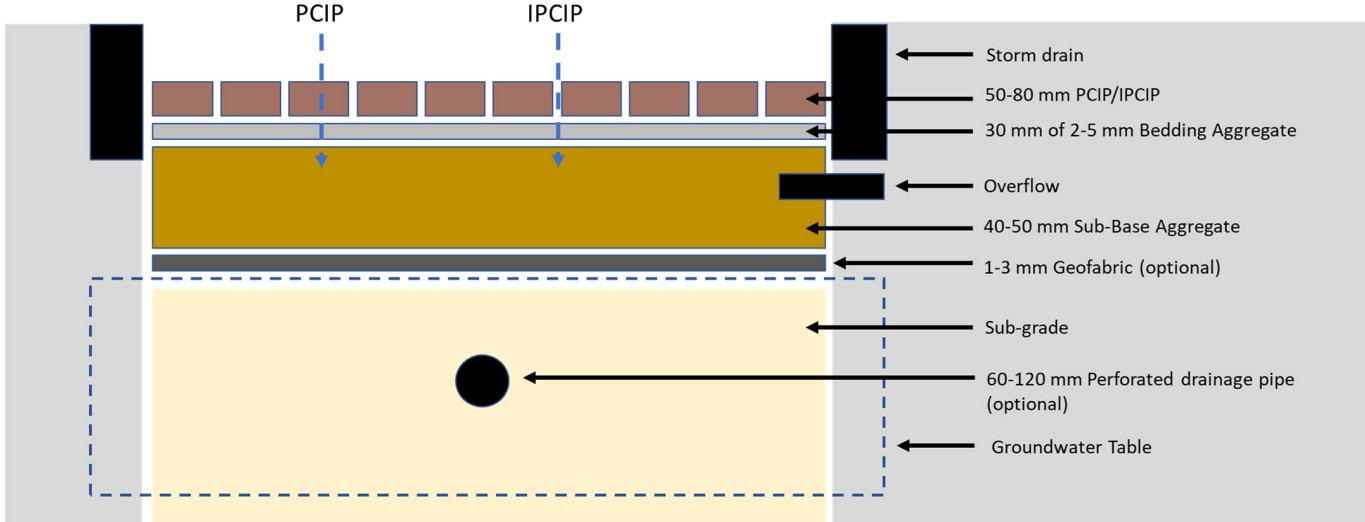

**Figure 1.** Typical setup of a Dutch infiltrating pavement structure (Adapted with permission from [28]).

The project Sponge City [29] aimed to tackle the knowledge gap by means of practice-oriented research. As part of this project, we carried out 81 full-scale infiltration tests (FSITs) [27] at 39 testing locations spread over 9 municipalities throughout the Netherlands. Using the results from these tests, we examined the functioning of infiltrating pavements in practice and investigated how system characteristics, environmental factors, and time influence the functioning in practice. At a selected sample of 16 testing locations, FSITs were performed before and after maintenance. The results of these FSITs were used to assess the effect of maintenance, as well as different maintenance methods, on the performance of infiltrating pavements in practice. Through this, we formulated an answer to the following research questions:

1. How do infiltrating pavements function in practice?
2. How do time, system characteristics, and environmental factors influence the performance of infiltrating pavements in practice?
3. To what extent can maintenance improve the performance of infiltrating pavements in practice?

In the paragraph Materials and Methods, we describe the methodological steps taken to collect and analyze the data. This concerns (1) setting up FSITs at the locations; (2) the registration of system characteristics, environmental factors, and maintenance per location; (3) the interpretation of FSIT results per test location; and (4) the analysis of the influence of time, system characteristics, environmental factors, and maintenance on the functioning of infiltrating pavements in practice.

The paragraph Results starts with a general presentation of the functioning of infiltrating pavements in practice. Next, results focusing on the influence of time, system characteristics, and environmental factors, as well as the effectiveness of maintenance, are discussed in more detail. In the Discussion, the main results are considered in light of previously conducted (scientific) research. In addition, we discuss the main limitations of the research executed. We end with a number of recommendations for policy, practice, and further research. The most important outcomes of the research are summarized in the Conclusion.

## 2. Materials and Methods

Within this research, we tested the functioning of infiltrating pavements in practice. For this, by means of 81 FSITs at 39 testing locations spread over 9 municipalities in The Netherlands, we investigated the infiltration rate of different forms of infiltrating pavements (PCIP, IPCIP). We determined the influence of time, system, and environmental factors on the infiltration rate by plotting the results of the full-scale infiltration tests (FSITs) against these factors. In a selected sample of 16 testing locations, executed in 7 municipalities, we examined the effect of different types of maintenance by conducting FSITs in these locations before and after maintenance. Nomination of these locations was performed with the municipalities participating in the project Sponge City. The following paragraphs describe the methodological steps for carrying out the FSITs and analyzing the results.

### 2.1. Test Set-Up at Location

Various methods are available to measure the effectiveness of infiltrating pavements in practice: Single- or double-ring infiltrometers [30,31], a full-scale infiltration test [27], or the use of a rainfall-simulator [10,32]. The experiments within this research were carried out using the full-scale infiltration testing methodology (FSIT, [27]). Previous research shows that experiments using the FSIT methodology yield more robust results than tests using a double-ring infiltrometer [27], because a larger part of the street surface is tested, which reduces the risk of outcomes that are not representative of reality [27].

According to the FSIT methodology [27], a representative section of the street with a size of at least 4 m$^2$ per location is demarcated with sandbags and subsequently flooded several times to a depth of at least 5 cm. Due to the demarcation with sandbags, the water cannot flow out to other drainage options. As a result, the rate of lowering of the water level within the testing plot provides a good representation of the infiltration rate (mm/h) at the location. The lowering of the water level was measured using three self-logging hydraulic pressure transductors (divers), applied with a measurement interval of 5 s. Manual measurements (applied with a measurement interval of 30 s) were used to control and back up the results of the self-logging hydraulic pressure transductors. Figure 2 shows a typical test setup to perform an FSIT on infiltrating pavements.

At 17 individual testing locations, the FSIT was repeated 2–3 times with a maximum time interval of 30 min in order to capture the infiltration rate under unsaturated and saturated conditions [28]. By conducting the FSIT before and after maintenance (performed at 16 individual testing locations, while accounting for the potential saturation conditions in place), the effect of maintenance on the performance of infiltrating pavements was quantified [19,28,32].

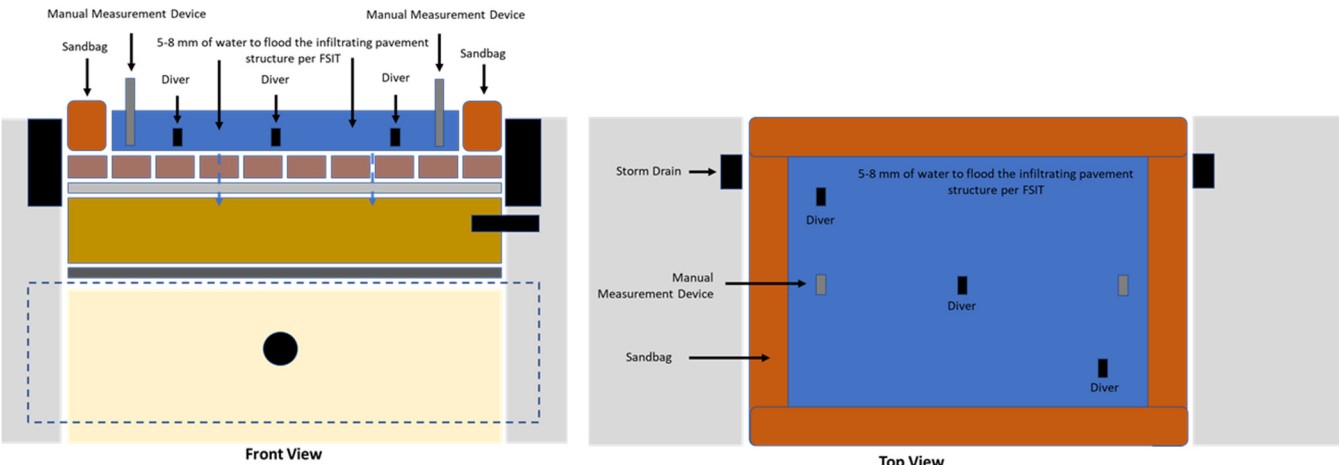

**Figure 2.** Typical setup of a full-scale infiltration test on infiltrating pavements (Adapted with permission from [28]).

### 2.2. Interpretation of Results per Location

Upon the field collection of the FSIT data, the infiltration rate of the infiltrating pavements (mm/h) was derived per FSIT from the average speed at which the water level dropped [27]. To do so, (1) a linear trend line was plotted along the curve showing the decrease in water level for each of the three self-logging hydraulic pressure transductors; (2) the slope and reliability ($R^2$) of this trendline were determined per self-logging hydraulic pressure transductor; (3) the weighted mean was taken of the results of the three hydraulic pressure water level meters; and (4) this weighted mean was finally extrapolated to arrive at a value of mm/h.

Using the derived infiltration rates under saturated and unsaturated conditions, as well as before and after the maintenance of the testing location, we quantified for each individual test location where these experiments were executed and what effect saturation, as well as the application of one or more maintenance methods, had on the infiltration rate of infiltrating pavements in practice.

### 2.3. Collection of System Characteristics, Environmental Factors, and Maintenance Type

Various factors can influence the infiltration rate of infiltrating pavements in practice, such as age, system characteristics (e.g., type of infiltrating pavement, road structure, groundwater levels, soil characteristics) and environmental factors (e.g., trees, shrubs, and traffic intensity) [14–21]. However, the possible influence of these factors in practice has been verified only up to a limited manner in practice and over longer periods of time using a large dataset [16]. To do so in this research, the following environmental factors were recorded for each measurement location and included in the measurement protocol: The presence of trees and shrubs; condition of the paving (rutting, plate formation, loose stones); neighborhood typology; traffic and parking intensity; and subsoil (groundwater-level, soil characteristics). In addition to the environmental factors, some characteristics of the infiltration system have also been determined per location: The structure and condition of the joints; construction of the street layer; construction of the foundation; angle of the surface; the presence of storm drains; and system operations. Moreover, the age and type of pavement (PCIP, IPCIP) were captured. Supplementary Table S1 summarizes the factors and characteristics recorded for each testing location. The table also describes the maintenance methods that were investigated: Regular sweeping, vacuum-cleaning, high-pressure air, and the replacement of the joint-filling material.

### 2.4. Overarching Analysis of the Results

For the entire set of test results (Supplementary Table S2), we assessed the performance of infiltrating pavements in practice. For this, we first compared the collected infiltration

rates with Dutch guidelines (with 60 mm/h being representative for a 100 year-1 h storm event, [33]) and international standards (194 mm/h, [34]). Secondly, we examined the statistical relationship between the performance of infiltrating pavements and time (age) by using a linear and exponential regression analysis. Thirdly, we evaluated the influence of system characteristics and environmental factors on the (change in) infiltration rate over time.

The effect of (different types of) maintenance was tested by comparing the infiltration rates before and after maintenance had taken place. In addition, a distinction was made between the effect of different cleaning methods, the effect of cleaning on different types of permeable pavement, and changes in the effect of cleaning as a function of the lifespan of the permeable pavement.

## 3. Results

### 3.1. System and Environmental Characteristics of Infiltrating Pavements

Of the 39 testing locations included in this study, 18 locations were constructed with a PCIP pavement and 21 locations with an IPCIP pavement. At a number of testing locations, storm drains were installed to obtain the water from the street into the subsoil. The FSITs were always set up and carried out in such a way that these storm drains did not play a role in the measured infiltration capacity.

The age of the testing locations included in this study is, on average, 6.3 years and varies between 0 and 13 years. Most testing locations have a one-sided cant or are barrel-round. A smaller proportion of the tests were carried out at locations with a roof, flat, or hollow street profile. The joint width of the pavement at the various test locations is up to 8 mm wide. The joint material commonly used is Brekerzand 0–2, Basalt split 0–2, and Bestone split 1–3. The joint condition is reported as good at most testing locations, although sometimes polluted or overgrown with moss. Bestone Split, Hollands Steenslag, or crushed stone 2–6 is used as paving material in most testing locations. Aquabase C1 1–3 has been used in a few places. The thickness of the street layer varies between 3 and 7 cm. The foundation thickness varies between 25 and 35 cm, and the material used for the foundation is usually mixed granulate or crushed stone 3/32 to 8/32. In some places, Drainmix 4/16 or Aquabase A5 8/32 was used. Individual system characteristics per location are described in Supplementary Table S2.

The environmental factors of the 39 testing locations differ. Most FSITs were carried out in neighborhoods of the 'sub-urban expansion' type. In addition, several infiltration tests were carried out in 'residential areas', 'renewed area', 'post-war residential areas' and 'post-war low-rise garden city', and 'industrial estates'. The presence of trees was reported near 35 testing locations, while shrubs were present at 32 testing locations. The testing locations generally score low to average for traffic and parking intensity.

The 16 testing locations (9 IPCIP locations, 7 PCIP locations) where maintenance tests were carried out as part of this study are, on average, 7.2 years old and vary in age from 0–12 years. The presence of trees was reported for 14 of these testing locations, while shrubs were reported at 13 locations.

### 3.2. Infiltration Rates of Infiltrating Pavements and (Inter)National Standards

Measured infiltration rates under uncleaned and unsaturated conditions vary from 35 to 1719 mm/h (IPCIP: 35–1186 mm/h, PCIP: 57–1719 mm/h) (Figure 3). At the 39 testing locations where FSITs have been carried out under uncleaned and unsaturated conditions, we find that the measured infiltration rate, with an average of 364 mm/h (median 167 mm/h), is above the Dutch and international reference values of 60 and 194 mm/h, respectively [33,34]. An infiltration rate of at least 60 mm/h was measured at 87.2% of the testing locations under uncleaned and unsaturated conditions. At 41% of the testing locations, the measured infiltrating rate was higher than 194 mm/h. However, in most locations, the area connected to the permeable pavement installation is higher than the installation alone, therefor 60 mm/h is the minimum and should be higher in practice. Neither the

observed infiltration rates nor the Dutch or International infiltration standards have been corrected for this difference in the area connected to the permeable pavement installation.

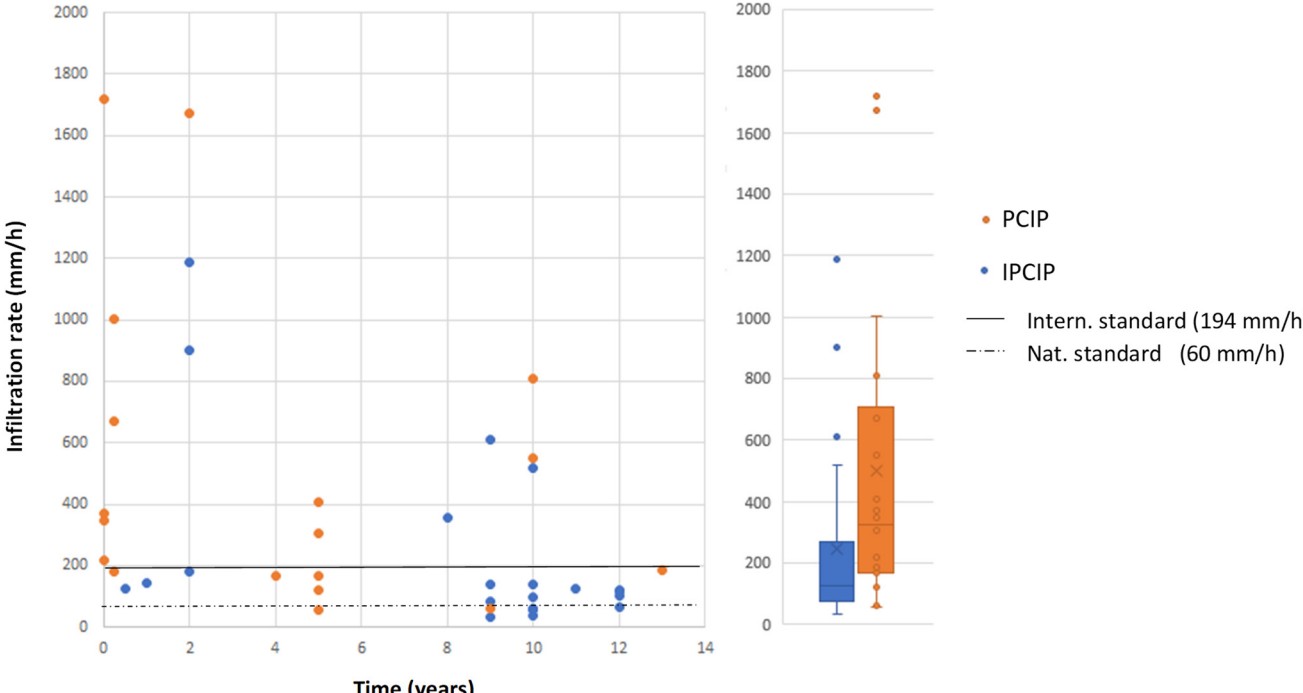

**Figure 3.** Functioning of infiltrating pavements in practice. Figure 1a shows the infiltration rate of PCIP (orange) and IPCIP (blue) systems under uncleaned and unsaturated conditions versus time. Solid and hatched lines indicate the international and national standards. Figure 1b shows the spread (mean (visualized by the X) +- 1 standard deviation in the box and whiskers) in infiltration read per PCIP (orange) and IPCIP (blue) system. Individuals observations are visualized by the dots.

The measured infiltration rates for IPCIP and PCIP differ substantially. IPCIP present an average infiltration rate of 248 mm/h (median: 123 mm/h), in contrast to the average infiltration rate of 500 mm/h (median: 326 mm/h) found for PCIP. A total of 85.7% and 88.9% of the investigated IPCIP and PCIP locations, respectively, meet the Dutch reference value of 60 mm/h. The international standard of 194 mm/h is met by 23.8% and 61.1% of the IPCIP and PCIP locations, respectively.

The difference in age of the two infiltrating pavement types can be seen as one explanatory factor for the observed variations in the infiltration rate between IPCIP and PCIP. The average age of the IPCIP locations investigated is 8.1 years (median: 10 years), whilst for PCIP locations investigated, the average age is 4.1 years (median: 4.5 years). While other system characteristics have also been initially considered as explanatory factors for the observed variation difference in performance (such as the structure and condition of the joints, the construction of the street layer and foundation, and ground-water levels and soil type), not enough data were available to assess this influence statistically.

### 3.3. Saturation Effects on Infiltrating Pavements

The results from saturation FSITs, executed at 17 testing locations under uncleaned conditions, show that saturation decreases the infiltration rate of infiltrating pavements by 34.6% on average (median: 36.3%) with infiltration rates ranging from 3.7% to 74.6% of the original values given under unsaturated and uncleaned conditions (Figure 4). As such, the saturation of the system caused one testing location to not be able to meet the international standard, whilst two testing locations were not able to meet the national standards. All of them are IPCIP systems.

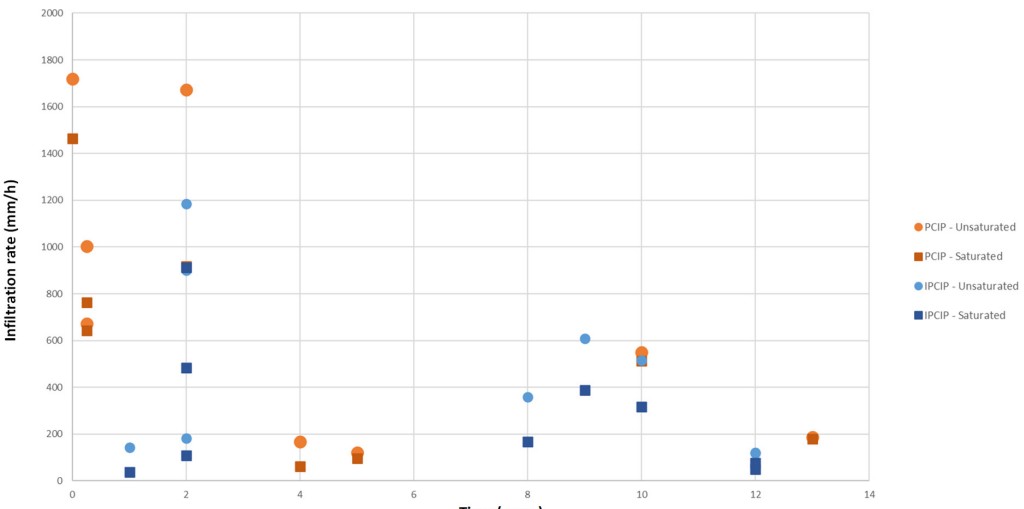

**Figure 4.** Infiltration rate of PCIP (orange, red) and IPCIP (light, dark blue) systems under unsaturated (circles) and saturated (squares) conditions.

Saturation effects differ between PCIP and IPCIP. Whilst saturation decreases the infiltration rate of PCIP by 22.9% on average (median: 17.9%), for IPCIP, an average decrease in infiltration rates of 45.1% due to saturation was found (median: 40.7%). An explanation for this difference could be the age difference between PCIP and IPCIP systems, as well as the differences in system characteristics: In comparison to IPCIP systems, the porous pavers of PCIP systems provide the extra storage capacity that may be utilized optimally under saturated conditions.

### 3.4. Effect of Time, System Characteristics, and Environmental Factors on Infiltrating Pavements

Results from FSIT executed at 39 testing locations show that the age of the infiltrating pavements is negatively correlated with the measured infiltration rate under unsaturated and uncleaned conditions (Figure 5). That is, the older the system, the lower the infiltration rate. The infiltration rate of all infiltrating pavements ($n$ = 39) decreases yearly with 41.8 mm/ h on average, whilst values for PCIP ($n$ = 18) and IPCIP ($n$ = 21) do not differ significantly from each other (36.2–36.6 mm/h/y). However, the $R^2$ value, which represents the explanatory power of this correlation, is not strong (all systems: 0.19, PCIP: 0.09; IPCIP: 0.22). This indicates that, in addition to age, other factors also have a substantial influence on the infiltration rate of a specific test location, for example, environmental factors.

Whilst these environmental factors (e.g., traffic and parking intensity, presence of shrubs and trees) have been considered as an additional explanatory factor for the observed decrease in infiltration rate through time, we did not find a statistically significant influence.

Application of the observed time effects on the infiltration rate of infiltrating pavements (Table 1) shows that after a time period of 5 years from the moment when the experiments were carried out, only 23.8% (19%) to 55.6% (38.9%) of the testing locations would still meet the national (international) standards for IPCIP and PCIP systems, respectively. After 30 years, only 4.8% (0%) to 11.1% (11.1%) of the testing locations would still meet the national (international) standards for IPCIP and PCIP systems, respectively. Using the time effects found for PCIP and IPCIP systems results in an estimated average lifespan of 16.5–13.0 and 13.8–10.2 years using the national (60 mm/h) and international standards (194 mm/h) standards, respectively.

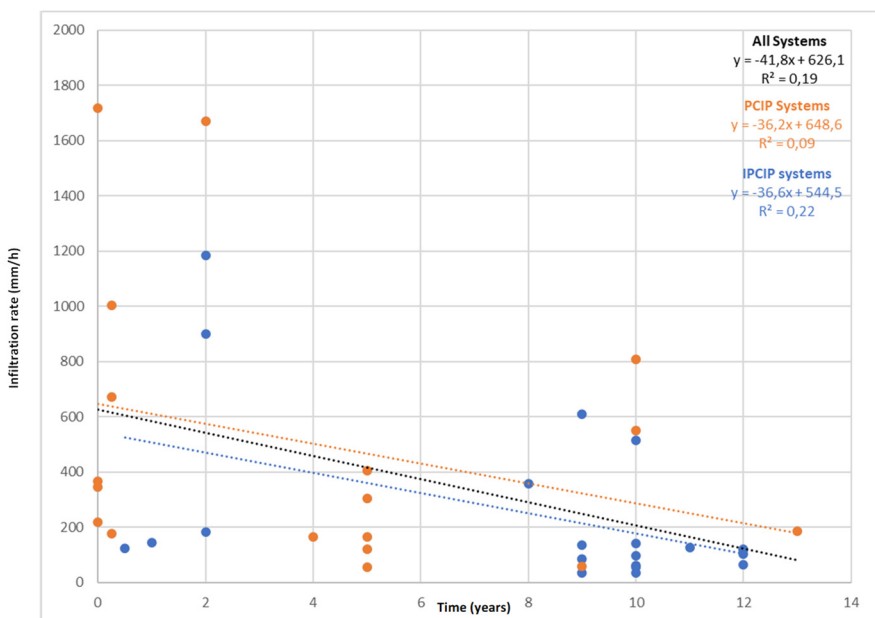

**Figure 5.** Linear trend lines indicating the decay in infiltration capacity of all (black), PCIP (orange), and IPCIP (blue) systems over time. The given equations and $R^2$ of the linear trend lines indicate the decay in infiltration rate in mm/h per year and the explanatory power of time/age (goodness-of-fit) to the yearly decay in infiltration rate. The low values of $R^2$ indicate that the predictive value of the equation is not strong.

**Table 1.** Application of the observed time effects on the infiltration rate of infiltrating pavements in relation to the national guidelines and international standards and percentage of systems (%) functioning according to (inter)national standards in years after the moment of measurement.

|  |  | Av. Lifespan | 0 yrs | 5 yrs | 10 yrs | 15 yrs | 20 yrs | 25 yrs | 30 yrs |
|---|---|---|---|---|---|---|---|---|---|
| National Standards | PCIP | 16.5 years | 88.9% | 55.6% | 33.3% | 27.8% | 22.2% | 16.7% | 11.1% |
| | IPCIP | 13.8 years | 85.7% | 23.8% | 19.0% | 14.3% | 9.5% | 4.8% | 4.8% |
| International standards | PCIP | 13.0 years | 61.1% | 38.9% | 27.8% | 22.2% | 16.7% | 11.1% | 11.1% |
| | IPCIP | 10.2 years | 23.8% | 19.0% | 14.3% | 9.5% | 4.8% | 4.8% | 0% |

*3.5. Effect of Maintenance on Infiltrating Pavements*

At 16 testing locations, FSITs were carried out to test the effectiveness of (different) maintenance methods on the infiltration rate of infiltrating pavement, including the regular use of a sweeper, the application of a vacuum cleaner, usage of a high-pressure air cleaner, and replacement of the joint filling material (Figure 6). The measurements show that the maintenance of infiltrating pavements generally leads to an increase in the infiltration rate. In three cases, maintenance resulted in a lower infiltration rate, all due to explainable reasons (replacement with wrong joint filling, cleaning under unfavorable—wet—circumstances, and regular street sweeping). On average, cleaning appeared to increase the infiltration rate by +306.2% (all systems, PCIP: +294.7%, IPCIP: +316.1%). The change in infiltration rate after cleaning varies from −30.2% (high-pressure air, when applied during wet conditions) to +1390% (vacuum cleaning). At one test location, the effect of replacing the joint filling of IPCIP with basalt and crusher sand was tested, with varying results. Where replacement of the joint filling with basalt results in a 252.1% increase in the infiltration rate, replacing the joint filling with crusher sand led to a slight decrease in the infiltration rate (−32.4%). Based on our results, the use of a high-pressure air cleaner seems to be the most effective for IPCIP. If performed properly (i.e., not under wet conditions), this cleaning methodology is able to clean the joints of IPCIP, which results in an increase

in the infiltration rate of, on average, 441.4% (−30% to +840%). When applied under wet conditions, the high-pressure air cleaner is not able to remove dirt from the joints, as shown by the −30.2% change in the infiltration rate for that specific testing location. The use of a high-pressure air purifier leads to an increase in the infiltration rate of, on average, 124.2% (93–155.3%) when applied to PCIP. While a high-pressure air cleaner is able to remove some dirt from the surface of infiltrating porous pavers, Figure 6 shows that the use of a vacuum cleaner is more effective for PCIP. Using a vacuum cleaner on PCIP results in an increase in the infiltration rate of 510.9% (38.2–1390.9%). Application of a vacuum cleaner on IPCIP leads to an increase in the infiltration rate of 149.1% (22.4–287.8%). No significant influence of time was found when comparing the effectiveness of the different maintenance techniques and PCIP/IPCIP systems against their age.

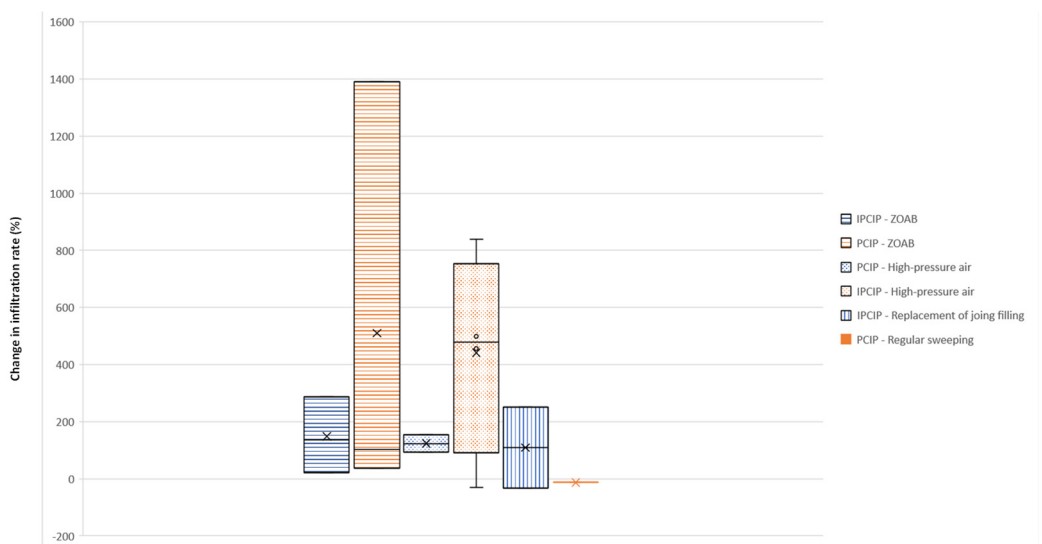

**Figure 6.** Boxplot showing the percentage change in infiltration capacity after maintenance, making a distinction in the effect of different maintenance techniques and IPCIP and PCIP systems. X visualizes the mean value.

The application of regular street sweeping as a maintenance methodology for infiltrating pavement was only tested once in this study with a negative result (PCIP: −12.8%). Although this result needs to be verified by more targeted maintenance tests using regular street sweeping, this result seems to be supported by the average decay in the infiltration rate over time. In other words, regular cleaning does not seem to be able to counteract the average decay in the infiltration rate over time, at most diminishing it by some percentages.

Application of the maintenance results found to fit PCIP (+510.9% using the Vacuum-cleaner) and IPCIP (+441.4% using the high-pressure air cleaner) systems best, regarding the yearly infiltration rates and expected lifespans, shows that, on average, carrying out maintenance 3.5–4.2 times (i.e., once every 9.8 or 7.1 years) is needed to extend the lifespan of PCIP and IPCIP to the desired 30 years given the national standards (60 mm/h). Using the same percentual increase in infiltration rate due to maintenance in combination with the international standards (194 mm/h), we find that, on average, carrying out maintenance 6.1 (once every 4.9 years)–7.1 (once every 4.3 years) times over the course of 30 years for IPCIP and PCIP systems, respectively, is needed to keep the systems functioning.

## 4. Discussion

### 4.1. Comparison of the Results with Earlier Research

Previous studies conducted on the infiltration rate of PCIP or IPCIP systems in the Netherlands and abroad present similar results as found in the former chapter [14–16,19–21,26,28,32], both in terms of the measured infiltration rate and the decline in infiltration rate in relation to the age of systems. Boogaard et al. [26,28] present

infiltration rates ranging from 21 mm/h to 503 mm/h, with an average of 150 mm/h. The average measured infiltration rate in this study is higher with an average of 364 mm/h. This difference is caused by the fact that the current study includes tests with relatively young and innovative forms of PCIP and IPCIP with a high to very high infiltration rate.

Based on an exponential decay function, Borgwardt [32], Boogaard et al. [26], and Winston et al. [19] conclude that the age of the system is negatively linked to the measured infiltration rate. Borgwardt [32] arrived at a decline of 10–25% of the infiltration rate over a period of 12 years, a conclusion supported by Boogaard et al. [26] and Winston et al. [19]. Based on a linear fit, our results present a relatively stronger decay over time (42 mm/year, i.e., 5.6–6.7% per year or 60–80% in 12 years). When applying an exponential decay function to our results—which does not necessarily result in a significantly higher $R^2$ in comparison to the linear fit—the results of our study show even higher decay values: 59.3–74.2% over a period of 12 years. Several factors may explain the observed difference in decay. For example, while Borgwardt [32] does not distinguish between locations that have been maintained or not, we only include locations that have not been cleaned (other than regular street sweeping) in our study for this analysis. Variations in the system characteristics and environmental factors of the different locations tested may also play a role in this. Finally, it is important that the dataset currently used is larger than the studies presented earlier ($n = 8$–55).

The relatively low $R^2$ associated with the trends found for the decay of the infiltration capacity of permeable pavements over time implies that age is only one explanatory variable for the measured decline in the infiltration rate. Other environmental factors and system characteristics also play a role. While various environmental characteristics are documented, insufficient data were available to produce significant statistics on this matter and more data are needed to underpin these findings.

While the studies mentioned above also hint at both environmental and climatological factors as a factor in the decay of permeable pavements over time—mostly due to clogging— Kia et al. [16] performed a systematic review of these clogging mechanisms. The retention and accumulation of solid particles, i.e., tree litter or sediments, is identified as the most prominent factor causing the clogging of permeable pavements over time [16]. These particles, usually broken down further by traffic—exacerbating the problem—fill and block void spaces, enhancing the further accumulation of fines [16]. While both clay and sand sediments can cause clogging, it is the particle size distribution of the clogging material in relation to the pore structure of the system that significantly influences the speed of clogging [14,15]. Upon drying, the accumulated particles form a hard crust that seals the voids [35]. Continuous wetting without dry periods also significantly affects the performance of permeable pavements [21]. Biological growth, e.g., due to street tree litter [36], occurs faster in continuous wet conditions and is believed to cause an earlier onset of clogging [37,38]. An advantage of decomposing leaves in comparison to other litter is, however, the elevated nutrient levels that stimulate microbial activity to enhance an accelerated hydrocarbon breakdown [39].

The 17 saturation tests performed at different locations show that saturation of the system leads, on average, to a decrease in the infiltration rate of 34.6% (median: 36.3%). This is an important notion since it shows that storm-after-storm events have a significant influence on the infiltration capacity of infiltrating pavements. The results from this study are consistent with those previously presented by Boogaard et al. [28]. Based on the results of infiltration tests conducted at 16 locations in the Netherlands, Boogaard et al. [28] calculated that saturation leads to a decrease in the infiltration rate from 23% to 65%.

The effectiveness of various maintenance techniques for restoring the infiltration capacity of clogged permeable pavements has been investigated in a number of studies with varying findings between different and even within a single study [16]. In a maintenance study similar to ours, Winston et al. [19], based on measurements at 10 different locations in the U.S. and Sweden (with PCIP and IPCIP), studied the effect of different maintenance techniques on the infiltration rate of systems with variable ages. Techniques studied in

that study were street sweeping, high-pressure air, high-pressure water, and milling. The results underline the importance of thorough cleaning for increasing the infiltration rate. Although the different cleaning techniques are not 100% effective, they generally show an increase in the infiltration rate. Street sweeping alone does not lead to an increase in the infiltration rate in a number of cases. This is in line with our result. Winston et al. [19] show that for PCIP, high-pressure air, high-pressure water, and milling, in particular, have a positive effect on the infiltration rate, a result that is in line with our research. According to Winston et al. [19], the removal of the top 2 cm of contamination from the joints of IPCIP did not lead to a significant improvement in the infiltration rate. Winston et al. [19] mainly attribute this result to the wrong choice of joint material at these locations, something also underpinned by Deo et al. [15] and Kia et al. [16]. These observations not only underline the importance of making the right choice when it comes to the management and maintenance of PCIP and IPCIP, but they also emphasize, once again, the importance of the correct design and choices in construction for the success of infiltrating pavements in practice [16].

### 4.2. Limitations of this Study and Recommendations for Further Research

When interpreting the results of this study, a number of assumptions, choices, and limitations in the conducted research must be taken into account. In the design and execution of the infiltration tests on location, the aim was to achieve the most consistent design and approach with regard to setting up the test sections, performing the measurements, and analyzing the results.

The testing of 81 FSITs spread over 39 testing locations was a combined effort of many people. However, this total appeared to still be too small a sample, especially when a breakdown is made over various systems, environmental factors, or maintenance strategies. In some cases, this leaves only a few measurements per category on which averages and statistics cannot always be based. Supplementing the dataset with additional FSITs, especially with environmental factors or system characteristics that are now relatively underrepresented in the dataset, is necessary to improve the robustness of the presented results.

When performing the analyses on the infiltration rate, the effect of environmental factors, and maintenance on this infiltration rate, as well as the decline of the infiltration rate over time, all FSITs were analyzed together and considered as one/equally interpretable. When interpreting the results, it should be taken into account, however, that different locations have been dimensioned in different ways. In most locations, for example, the area connected to the permeable pavement installation is higher than the installation alone. Observed infiltration rates have not been corrected for this difference in the area connected to the permeable pavement installation. One should therefore be careful when comparing absolute values from one location to another. However, by comparing the group averages, we can provide the first indication of the role of time, system characteristics, environmental factors, and maintenance in the functioning of infiltrating pavements in practice. Supplementing the dataset with information about the dimensioned infiltration rate during the design/construction of the individual testing locations and expressing the decline in infiltration rate as a percentage of the original design value versus time would be the next step to better express the functioning of infiltrating pavements in practice. With that extra information, one would be able to compare the infiltration time per location in a more objective manner, as well as the potential lifetime and the influence of environmental factors and maintenance [15,16]. Municipalities, however, have not yet always archived this information in a consistent manner readily available for analysis.

This study on infiltration rates of infiltrating pavements in practice was carried out using FSITs with a test area of approximately 4 m². Scientific research shows that tests using an FSIT yield more robust results than tests using a double-ring infiltrometer [27]. After all, a larger part of the street surface is tested, which reduces the risk of outcomes that are not representative of the actual situation. However, even with a test area of 4 m², there is still a chance of test results that deviate from the actual situation. For more certain

insight into the infiltration capacity, it is therefore recommended to perform a full-scale test in which the entire street surface is included in the test.

The three self-logging hydraulic pressure transductors used in this study have not been individually calibrated for each testing location. In some cases, this resulted in negative measured water levels. This was not a problem as these absolute values of water heights are not used to calculate the infiltration rate, but rather the decrease in water height over time. The lack of a calibration process per measuring location for the divers therefore has no influence on the calculated infiltration rate. In line with previously published research, the infiltration rate in mm/min and mm/h is calculated by fitting a linear trendline and determining the slope of this trendline for the individual infiltration curves. As a check, the $R^2$ (goodness-of-fit) has been determined for all established trend lines. In general, this $R^2$ is good with a mean of 0.94, in line with previous studies [26–29]. However, choosing a different type of trendline—other than the linear fit applied here—can potentially lead to even further improvement of the $R^2$ and different results with regard to the measured infiltration rate. Further research is necessary for this.

This study examined the infiltration rate of infiltrating pavements and the effects of time, system characteristics, environmental factors, and maintenance on this infiltration rate. However, the functioning of an infiltration facility in practice is determined by the entire construction (including the storage capacity of the foundation construction in relation to the k-value of the subsoil, the presence/absence of drainage facilities such as sewers and gullies, etc.) and the dimensions of the draining surface for which the construction is designed for. In order to determine whether a presented solution at the neighborhood level meets the expectations and requirements, additional research is necessary for which all these aspects are taken into account. In order to investigate this, a new research program has been set up.

The results of this study show that maintenance in most cases has a large positive effect on the infiltration rate. At various locations, however, testing was carried out only once or several times before and after cleaning, and only one or a few cleaning methods were put into practice. Additional measurements and analysis of the effectiveness of other cleaning methods and the decrease in the infiltration rate over time before and after cleaning are therefore desirable. In order to guarantee the implementation of this type of multi-year measurement, we advise municipalities to develop monitoring protocols in which they record this. Only in this way can the longer-term effects of environmental factors and maintenance on infiltrating pavements actually be determined, providing a basis for the development of an optimal maintenance schedule to ensure the proper functioning of the infiltrating pavement, together with associated cost–benefit assessments to the added value of this type of climate adaptation.

## 5. Conclusions

Infiltrating pavements are potentially effective climate adaptation measures to counteract arising challenges related to flooding and drought in urban areas. In various Dutch municipalities, different forms of infiltrating pavements have been constructed over the past 10–12 years as an alternative to an urban stormwater drainage system in order to first retain and store the rainwater and only then drain/discharge it. In practice, however, these infiltrating pavements do not always function optimally, partly due to problems with the decrease in infiltration capacity and lack of clarity about management and maintenance.

As part of the project Sponge City, using 81 full-scale infiltration tests we investigated how infiltrating pavements function in practice. First, we examined the functioning of infiltrating pavements through time. In addition, the effect of system characteristics and environmental factors was studied. Finally, maintenance tests were performed to assess the effect of different maintenance methods on the infiltration rate.

The results of this study show that with an average infiltration rate of 364 mm/h, the investigated infiltrating pavements function well above the Dutch and international standards. An infiltration rate of at least 60 mm/h was measured at 87.2% of the testing

locations. In 41% of the testing locations, the measured infiltration rate was higher than 194 mm/h. The measured infiltration rates for IPCIP (248 mm/h) and PCIP (500 mm/h) systems differ substantially.

Infiltration rates of infiltrating pavements decrease over time, by 41.8 mm/h per year on average (PCIP: 36.2 mm/h; IPCIP: 36.6 mm/h). Application of the trend found for the decay of the infiltration rate over time shows, without maintenance other than regular street sweeping, estimated average lifespans of 13.0–16.5 and 10.2–13.8 years using the international (194 mm/h) and national (60 mm/h) standards for PCIP and IPCIP systems, respectively. The $R^2$ associated with these trends is, however, small. This implies that age is only one explanatory variable for the measured decline in infiltration rate. Other factors also play a role, such as system characteristics and environmental factors. Too little data were available, however, to quantify this effect. Expanding the dataset with testing locations targeting specific system characteristics and environmental factors, as well as exploring the possible cross relationships between these influencing factors, is needed to further specify these results.

Maintenance can play a major role in preserving or improving the performance of infiltrating pavements in practice. The analysis of the maintenance FSIT results, executed at 16 individual testing locations, shows that cleaning infiltrating pavements can increase the infiltration rate by 306.2% on average. Based on our results, the use of high-pressure air appears to be effective mainly for IPCIP (+441.4%), while the use of a vacuum cleaner achieves the best results for PCIP systems (+510.9%). Application of the found maintenance results for PCIP and IPCIP systems shows that, on average, maintenance is needed once every 7.1 to 9.8 years to extend the lifespan of PCIP and IPCIP to the desired 30 years given the Dutch national guidelines (60 mm/h).

The conclusions of this study are based on the results of 81 infiltration tests, performed at 39 individual testing locations spread out over 9 municipalities in the Netherlands, with infiltrating pavements of varying ages. They offer both municipalities and SMEs a good starting point to continue working on the application of infiltrating pavements in practice and provide the Dutch Delta Program handles to further promote the implementation of infiltrating pavements, if implemented and maintained well, as part of the ambition to make The Netherlands water-robust and climate-resilient by 2050. Moreover, the findings can be used by a broader audience including urban planners, policymakers, engineers, urban foresters, researchers, and the general public. However, only with multi-year measurements at individual locations following a strict monitoring protocol can the longer-term effects of environmental factors and maintenance on infiltrating pavements actually be determined, providing the basis for the development of an optimal maintenance schedule and associated cost–benefit assessments to the added value of this type of climate adaptation.

**Supplementary Materials:** The following supporting information can be downloaded at: https://www.mdpi.com/article/10.3390/w14132080/s1, Table S1: Database showing the registered metadata (system characteristics and environmental factors) per individual testing location. Table S2: Database containing all metadata and full-scale infiltration test results for individual testing locations incorporated in this study.

**Author Contributions:** Conceptualization: T.I.E.V., F.C.B. and J.K.; methodology: T.I.E.V., F.C.B. and J.K.; data collection: T.I.E.V. and F.C.B.; data analysis: T.I.E.V.; writing: T.I.E.V., F.C.B. and J.K. All authors have read and agreed to the published version of the manuscript.

**Funding:** This research was funded by Regieorgaan SIA, grant numbers RAAK.MKB.08.023 and RAAK.PUB07.012.

**Institutional Review Board Statement:** Not applicable.

**Informed Consent Statement:** Not applicable.

**Data Availability Statement:** See Supplementary Material.

**Acknowledgments:** We thank all (former) colleagues at HHG, HvA, and HR for their contribution to the field collection of the data, in particular Kylian Postema, Tom Schoenmaker, and Jonathan Lekkerkerk. We thank all municipalities for their willingness to host our field experiments.

**Conflicts of Interest:** The authors declare no conflict of interest.

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
