# Peer review of "Unlocking the Potential of Permeable Pavements in Practice: A Large-Scale Field Study of Performance Factors of Permeable Pavements in The Netherlands"

_water, doi:10.3390/w14132080_

Round 1

Reviewer 1 Report

This manuscript investigates the performance of the permeable pavement. However, the manuscript is not well organized and does not show enough results. Also, the results of the manuscript are not meaningful, and difficult to have an academic contribution. Therefore, I do not recommend that this manuscript is qualified for publication Water journal.

Author Response

We thank Reviewer #1 for the time taken to review our manuscript and have made some amendments to the manuscript to improve the clarity of methods and results. In the document attached we provide a point-per-point answer on the reviewer’s comments.

Reviewer 2 Report

A couple of formal errors need to be corrected, as references are included in the text instead of numbers (lines 133 and 420).
In section 1 of materials and methods, the characteristics of each FSIT should be better clarified in order to be able to relate them to the results obtained.  It would be very useful to include a typical section of the pilot plant.
What is the time period tested for the infiltration properties of the pavements? It would be necessary to clarify, after how long the pavements have been in operation, how long the maintenance of the pavements was carried out. 
When showing the data in the figures, it only indicates whether it is PCIP or IPCIP but there are more characteristics that differentiate these pavements that are not represented in the results, so it is difficult for the reader to verify the conclusions of the results.

Author Response

We thank Reviewer #2 for his/her time taken to review our manuscript and we are delighted that the reviewer thinks that our manuscript is fit for publication. In revising our manuscript we have taken the recommendations of Reviewer #2 into consideration. While doing so we think we have significantly improved our description of methods and results. Please find a point-per-point answer to the suggestions of Reviewer #2 in the attached document.

Reviewer 3 Report

Overall, this study is well designed and well presented. And the findings can be used by a broader audience including urban planners, policymakers, engineers, urban foresters, researchers, and the general public. Here are just some minor comments for the authors’ consideration:

-Ls80-93: these two paragraphs could be deleted.

-L101: Authors need to explain how 16 testing locations were selected out of the total of 39 testing locations.

-Ls466-477: these two paragraphs could be deleted.

-At the end of the conclusion, it might be helpful to provide some policy recommendations regarding the Dutch Delta Plan based on the findings.

Author Response

We thank Reviewer #3 for his/her positive evaluation of our manuscript and feel delighted that Reviewer #3 shares the feeling that our manuscript, and the results presented within it, can be of value for a broader audience. In revising our manuscript we have taken the recommendations of Reviewer #3 into consideration. While doing so we think we have significantly improved our description of methods. Please find a point-per-point answer to the suggestions of Reviewer #3 in the attached document.

Round 2

Reviewer 1 Report

The authors tried hard to revise the manuscript based on the reviewer's comments. I recommend the current manuscript is qualified to publish in the water journal